# Exploring Mitochondrial DNA Copy Number in Italian Children with ADHD: Implications for Neurobiological Mechanisms

**DOI:** 10.3390/diseases13110378

**Published:** 2025-11-19

**Authors:** Luigi Citrigno, Annamaria Cerantonio, Ludovico Neri, Pierluigi Sebastiani, Alessia Colanardi, Gabriele Turacchio, Tiziana Del Beato, Beatrice Marziani, Anna Aureli

**Affiliations:** 1CNR Institute for Biomedical Research and Innovation (IRIB), Loc. Burga, 87050 Cosenza, Italy; annamaria.cerantonio@irib.cnr.it; 2Neurology and Psychiatry Unit for Children and Adolescents, San Salvatore Hospital, Via L. Natali, 1, 67100 L’Aquila, Italy; ludovico.neri@graduate.univaq.it; 3CNR Institute of Translational Pharmacology, Via Carducci 32, 67100 L’Aquila, Italy; pierluigi.sebastiani@ift.cnr.it (P.S.); alessia.colanardi@ift.cnr.it (A.C.); gabriele.turacchio@ift.cnr.it (G.T.); tiziana.delbeato@ift.cnr.it (T.D.B.); 4Emergency Medicine Department, Sant’Anna University Hospital, Via A. Moro, 8, 44124 Ferrara, Italy; mrzbrc@unife.it

**Keywords:** ADHD, aggressive behavior, mtDNA-cn, MAOA, 5-HTT, polymorphisms

## Abstract

Background: Attention-deficit/hyperactivity disorder (ADHD) is a neurodevelopmental condition frequently accompanied by behavioral dysregulation. While genetic factors involving monoaminergic systems have been implicated, emerging evidence suggests a role for mitochondrial dysfunction in ADHD pathophysiology. Mitochondrial DNA copy number (mtDNA-cn), a surrogate marker of mitochondrial biogenesis and cellular energy demand, may reflect underlying neurobiological alterations and oxidative stress-related mechanisms relevant to ADHD. Methods: We assessed mtDNA-cn in the peripheral blood of 56 Italian children and adolescents with ADHD and 27 age- and sex-matched healthy controls. ADHD symptoms and aggressive behavior were evaluated using DSM-5 criteria and the Conners’ 3 Rating Scales. Genotyping was performed for *MAOA* (rs6323, rs1137070) and 5-HTT (rs4795541) polymorphisms. Results: ADHD patients showed significantly higher mtDNA-cn than controls (*p* = 0.002), supporting mitochondrial dysregulation. Comparing the ADHD patient subgroups with aggressive behavior and those without, a non-significant reduction in mtDNA-cn was observed in the first subgroup. Notably, individuals with the TT genotype (rs6323) or CC genotype (rs1137070) had significantly higher mtDNA-cn compared to controls with the same genotypes (*p* = 0.031). Similar increases were seen across all 5-HTT rs4795541 genotypes in ADHD patients. Conclusions: Our findings suggest that mitochondrial alterations may contribute to ADHD pathophysiology. The association between mtDNA-cn and monoaminergic gene variants highlights a potential link between neurotransmitter metabolism, oxidative stress, and mitochondrial function. Thus, mtDNA-cn may serve as a peripheral biomarker and therapeutic target in ADHD.

## 1. Introduction

Attention-deficit/hyperactivity disorder (ADHD) is a highly heterogeneous neurodevelopmental disorder characterized by persistent patterns of inattention, hyperactivity, and impulsivity and associated with cognitive, educational, and social dysfunction [1]. Also, aggressiveness is often observed in individuals with ADHD and can significantly impact social functioning and overall quality of life [2]. As reported in Shelton et al., the aggressiveness associated with hyperactive–impulsive–inattentive behaviors in children increases the risk of various psychological, academic, emotional, and social problems compared to either behavioral pattern observed alone [3]. Moreover, studies on aggressive, hyperactive–impulsive children suggested that impulsive aggression represents a high risk profile towards the development of adult antisocial behavior [4]. Historically considered a male-dominant condition with a 3:1 ratio of boys to girls [5], actually, in females, ADHD shows a different profile of symptoms and associated comorbidities, making it more difficult to be identified [6]. Indeed, females with ADHD exhibit greater intellectual impairments and internalization of problems than boys and lower ratings on hyperactivity, inattention, impulsivity, and externalizing problems [7]. The ADHD global prevalence in children and adolescents varies depending on age (7.6% in children aged 3 to 12 years and 5.6% in teenagers aged 12 to 18) [8], and it can also persist into adulthood (up to 85% of cases) [1]. Interestingly, ADHD symptoms tend to change across the lifespan, so children and adults with the disorder may exhibit very different phenotypes. Often, in adulthood, hyperactivity is replaced by restless feelings and discomfort, and inattention may manifest as difficulty in managing tasks or work activities [9]. Multiple factors, from genetic to environmental, can contribute to ADHD onset [10].

As regards the influence of genetics on disease onset, variations in genes associated with the neurotransmitter system, particularly those encoding the dopamine receptors (DRD2 and DRD4), the dopamine transporter (DAT1), the serotonin transporter protein (5-HTT), and monoamine oxidase (MAOA) enzymes, have been widely [11].

Twins and family studies evidence that genetic factors significantly increase vulnerability not only to ADHD but also to antisocial and aggressive behaviors in affected individuals [12,13]. Indeed, genes may modulate the functioning of brain circuits involved in impulse control, emotional regulation, and social behavior, thereby predisposing individuals to exhibit aggressive and antisocial tendencies [14]. In particular, rs6323 and rs1137070 polymorphisms, located in exons 8 and 14 of the *MAOA* gene, which influences monoamine metabolism, seem to be responsible for aggressiveness, impulsivity, and antisocial behavior observed in ADHD patients [15].

Also, the serotonin transporter gene (*SLC6A4*) encoding for the 5-HTT has been linked to impulsive-aggressive behavior. It is known that the 44-bp insertion/deletion variable number tandem repeat polymorphism (VNTR) (rs4795541) in the promoter region of the 5-HTT gene generates the long (l) and the short (s) allelic isoforms. Besides the difference in the transcriptional activity related to serotonin uptake (higher in l isoform and lower in s isoform), several studies underlined a remarkable association between these isoforms and impulsive–aggressive behavior traits [16,17].

Despite this, the biological mechanism underlying ADHD is not yet fully understood [18]. However, an increased oxidative stress has been described as one of the possible etiological factors in ADHD [19]. Indeed, it can cause cellular damage, DNA repair system malfunction, and mitochondrial dysfunction [19]. As is known, mitochondria are responsible for adenosine triphosphate (ATP) production and play a crucial role in cellular metabolism and energy homeostasis, particularly in the brain, which is highly energy dependent [20]. The complex role of mitochondria in ADHD is schematically shown in Figure 1. They have their own DNA (mtDNA), circular and double-stranded, that represents 1% of the total cellular one and contains 37 genes that are involved in mitochondrial and cellular functions [21]. Human cells contain from 1000 to 10,000 copies of mtDNA, and the mitochondrial DNA copy number (mtDNA-cn) serves as a key indicator of mitochondrial health and energetic capacity [22]. When mitochondria do not work properly, the resulting dysfunction leads to a series of changes ranging from inefficient cellular energy production and increased levels of reactive oxygen species (ROS) to the influence on the expression of nuclear genes involved in metabolism, growth, differentiation, and apoptosis [23]. All these changes may explain the role mitochondrial dysfunction plays in the pathogenesis of chronic and complex disorders [24].

Figure 1 displays the abundance of mitochondria in the brain and within synapses. Mitochondrial dysfunctions could cause changes in glycolytic pathways, ETC activity, and dysregulation of Na^+^/Ca^2+^ exchange. The endurance of oxidative stress and the breakdown of OXPHOS machinery exacerbate the unbalance between mitochondrial dynamics and mitophagy, leading to a depletion in ATP production and disruption of the integrity of mPTPs. All of these events trigger the NLRP3 inflammasome pathway and the concomitant release of DAMP molecules, thus contributing to the damaged brain networks noticed in ADHD. ETC: The Electron Transport Chain; OXPHOS: Oxidative phosphorylation; mPTPs: mitochondrial permeability transition pores; NLRP3: Nod-like receptor protein 3; DAMPs: Damage-Associated Molecular Patterns.

It has been suggested that mitochondrial dysfunction may be a susceptibility factor in the development of psychiatric diseases as schizophrenia, bipolar disorder, and depression [25,26]. Particularly, it has been proposed that in such disease processes, the accumulation of ROS and mtDNA mutations generates impaired cell functioning and replication, resulting in apoptosis and neuronal atrophy [27]. An involvement of mitochondrial dysfunction has also been indicated in ADHD, particularly regarding mtDNA-cn and its effects on neurochemical metabolism. Specifically, Öğütlü and colleagues, who first studied the relationship between mtDNA-cn and ADHD, found that the mtDNA-cn in children with ADHD was 1.3 times higher than that of normal controls [28]. Those findings have also been confirmed by others. Moreover, Payares et al., in their meta-analysis, provided an in-depth assessment of the mtDNA-cn role not only in ADHD development but also in that of autism spectrum disorder (ASD), which shares significant genetic overlaps with ADHD and often co-occurs with it. They have detected an increased mtDNA-cn in both conditions [29], emphasizing that mitochondrial dysfunction is implicated in these neurodevelopmental disorders and that these organelles may therefore be considered a potential target for effective therapeutic strategies [30]. However, further studies are needed to confirm the potential of mtDNA-cn as a biomarker for mitochondrial dysfunction in ADHD.

So, in order to deepen the understanding of the potential contribution of mitochondrial dysfunction to the etiology of ADHD, we provided an analysis of the changes in the mtDNA-cn in a group of Italian children with a diagnosis of ADHD.

## 2. Materials and Methods

### 2.1. Population Sample and Diagnostic Assessment

This study was based on a larger research project conducted in 2024, where *MAOA* and 5-HTT polymorphisms were investigated as susceptibility biomarkers of ADHD and/or aggressive traits in a cohort of 80 children and adolescents diagnosed with ADHD, recruited from the Unit of Child Neuropsychiatry at San Salvatore Hospital in L’Aquila, Italy. The age at diagnosis ranged from 6 to 18 years. ADHD diagnosis (inattentive, impulsive–hyperactive, combined) has been performed by trained psychiatrists who collected a complete medical history and evaluated the presence of symptoms of ADHD in different life settings, like home, school, and social ones. Psychiatric diagnosis has been confirmed according to the Diagnostic and Statistical Manual of Mental Disorders, Fifth Edition (DSM-V) and the International Classification of Mental and Behavioral Disorders, 11th revision (ICD-11).

The Conners’ 3 Rating Scales and the Strengths and Difficulties Questionnaire (SDQ) have been administered to the parents or caregivers with the purpose of better characterizing the symptoms of ADHD and the presence of aggressive behavior. For this latter purpose, we specifically focused on the Provocation/Aggressivity subscale, together with the interview validity indexes, according to the rules contained in the manual for the correct interpretation of the results. This comprehensive approach also enabled us to better analyze the presence of other neurodevelopmental and behavioral issues in the participants. Exclusion criteria were autism spectrum disorder, intellectual disability, or previously diagnosed genetic disorders. The control group comprised 80 age-, sex-, and ethnicity-matched healthy individuals. Before the recruitment, controls underwent the same clinical screenings performed in ADHD patients to rule out any subclinical ADHD symptoms or history of neuropsychiatric conditions.

For this study, participants with poor sample quality, missing mtDNA-cn data, or other incomplete datasets were excluded. A total of 56 children and adolescents with ADHD were included in the final analysis (Figure 2). Twenty-seven healthy controls were used for comparison.

The study was performed in accordance with the standards of the Ethics Committee (Code 0102550), and the written informed consent was obtained from all participants’ legal guardians.

### 2.2. Mitochondrial DNA Copy Number Quantification

Genomic DNA was isolated from peripheral blood (PB) cells using the QIAamp DNA Blood Mini Kit (Qiagen, Hilden, Germany) following the manufacturer’s instructions. After the extraction, the eluted DNA was stored at −20 °C until use. Purity and concentration of all DNA samples were assessed on a spectrophotometer (Beckman Instruments, Fullerton, CA, USA).

mtDNA-cn levels have been assessed by quantitative Real-Time PCR (qPCR) using the Absolute Human Mitochondrial DNA Copy Number Quantification qPCR Assay Kit (ScienCell, Carlsbad, CA, USA, AHMQ Cat. #8948).

The feature of this kit is to measure the average mtDNA-cn of samples using a set of primers that recognize and amplify a highly conserved mtDNA region. Simultaneously, the single copy reference (SCR) primers set recognizes and amplifies a small nuclear region within human chromosome 17, working as a housekeeping gene for data normalization. Both pairs of primers have been validated for qPCR by manufacturers through melt curve analysis, gel electrophoresis, and by template serial dilution to guarantee both amplification specificity and efficiency.

A reference genomic DNA with known mtDNA-cn, supplied with the kit, has been employed as a reference for calculating the absolute mtDNA-cn content of target samples.

qPCR amplifications, for reference DNA and for target samples, have been performed in two different reactions per sample using 10 µL of 2XGoldNStart TaqGreen qPCR master mix, 2 µL of primer solution (mtDNA or SCR), 7 µL of nuclease-free water, and 1 µL of target samples (5 ng) or reference DNA by using a StepOnePlus System (Applied Biosystem—Thermofisher Scientific, Waltham, MA USA). The thermal cycle conditions were carried out as follows: denaturation at 95 °C for 10 min; 32 cycles with denaturation at 95 °C for 20 s, hybridization at 52 °C for 20 s, and extension at 72 °C for 45 s.

### 2.3. Genotyping Analysis

Polymorphisms in *MAOA* rs6323, *MAOA* rs1137070, and 5-HTT rs4795541 were detected using high-resolution molecular biology techniques. Details of the methods are described in our previous work [31].

### 2.4. Statistical Analyses

As concerns mtDNA-cn evaluation, data have been analyzed using the Comparative ΔΔCq (Quantification Cycle Value) method. For target samples and for reference DNA, we obtained Quantification Cycle (Cq) values for both mtDNA and SCR.

For mtDNA, ΔCq was the quantification cycle number difference of mtDNA between the target sample and reference genomic DNA (ΔCq mtDNA = Cq mtDNA target sample − Cq mtDNA reference DNA). For SCR, ΔCq SCR was the quantification cycle number difference of SCR between the target sample and the reference genomic DNA (ΔCq SCR = Cq SCR target sample − Cq SCR reference DNA). ΔΔCq was obtained by the formula: ΔCq mtDNA − ΔCq SCR, with the purpose of determining the ratio of nuclear DNA versus mtDNA. The relative estimation of mtDNA-cn in each sample was calculated using the 2-ΔΔCq quantification method. The absolute mtDNA-cn of the target samples per diploid cell was extrapolated using the formula: reference sample mtDNA copy number × 2-ΔΔCq of each sample. Statistical analyses were performed using the R statistical software program (version 4.2.2).

Comparison between groups in non-normally distributed data was assessed using the Mann–Whitney U-test and the Kruskal–Wallis test with Dunn’s multiple comparisons test.

Statistical significance of differences between and within groups was determined by mean values of two-way ANOVA, followed by the Bonferroni multiple comparisons test. Statistical significance was set at *p* < 0.05. All the figures were generated with BioRender.com.

## 3. Results

This case–control study examined 56 children with ADHD (54 males and 2 females, with a mean age of 10.1 ± 2.6 years). According to DSM-V criteria, the subtype distribution was the following: 5.4% hyperactive–impulsive, 16% predominantly inattentive, and 78.6% combined subtype. Patients were further classified into two subgroups based on aggressive behavior presence (n = 32, 57%) or its absence (n = 24, 43%). The demographic and clinical characteristics of ADHD children are displayed in Table 1. Twenty-seven age- and sex-matched healthy individuals were used for comparison.

Firstly, we compared mtDNA-cn between the control and ADHD group. The assessment of mtDNA-cn in the peripheral blood of our cohort showed a statistically significant increase of this parameter in ADHD patients compared to healthy controls (mean ± SD: 1.51 ± 2.7 vs. 0.46 ± 0.46) (*p* = 0.002) (Figure 3), implying that mitochondrial dysregulation may contribute to ADHD pathology.

Moreover, considering that aggressive behavior is a common feature among individuals with ADHD, we examined mtDNA-cn levels based on aggressiveness score (aggressive ADHD and non-aggressive ADHD). Our findings indicated a decrease in mtDNA-cn levels in the subgroup of aggressive ADHD patients compared to non-aggressive ADHD patients (mean ± SD: 1.24 ± 1.8 vs. 2.26 ± 4.0). Although the difference did not reach statistical significance, mtDNA-cn might be negatively associated with aggressive behavior in ADHD patients (Figure 4).

Additionally, we investigated the relationship between mtDNA-cn and genotypes of MAOA and 5-HTT. Results of rs6323 *MAOA*, rs1137070 *MAOA*, and 5-HTT rs4795541 allele (af) and genotype frequencies found in ADHD patients and controls have been extrapolated from our previous work and are listed in Table 2.

Differences in mtDNA copy number between genotypes in *MAOA* and 5-HTT polymorphisms are shown in Figure 5 and Figure 6. In detail, we found that the mtDNA-cn in the TT genotype in rs6323 was significantly increased when compared with that of the controls (mean ± SD: 1.41 ± 1.8 vs. 0.36 ± 0.2) (*p* = 0.031, Figure 5a). Similarly, the mtDNA-cn in the CC genotype in rs1137070 was statistically higher than that of the controls (mean ± SD: 1.41 ± 1.8 vs. 0.36 ± 0.2) (*p* = 0.031, Figure 5b). Regarding the association between rs4795541 in 5-HTT and mtDNA-cn, we highlighted that the mtDNA-cn in all genotypes (LL, SS, and LS) was statistically higher than that observed in the control group (mean ± SD: 2.78 ± 5.0 vs. 0.40 ± 0.3; 2.62 ± 3.1 vs. 0.46 ± 0.2; 1.06 ± 1.1 vs. 0.49 ± 0.5) (*p* = 0.034, *p* = 0.017, and *p* = 0.02, respectively, Figure 6a–c).

We next stratified aggressive ADHD patients on the basis of polymorphisms in *MAOA* and 5-HTT genes, with the purpose of identifying mechanisms underpinning the hypothesis that mtDNA-cn fluctuations could reflect the aggressive attitudes determined by different genotypes. We did not find any significant association between mtDNA-cn levels and aggressive behavior in ADHD patients carrying different rs6323 and rs1137070 *MAOA* genotypes. Quite the opposite, we observed a statistically significant association between mtDNA-cn content and aggressive ADHD patients homozygous (SS) and heterozygous (LS) for 5-HTT rs4795541 genotype variants (mean ± SD: 3.1 ± 3.9 vs. 0.5 ± 0.32) (Figure 7).

## 4. Discussion

In the present preliminary study, we observed a significant increase in mtDNA-cn levels in our cohort of ADHD patients compared to healthy controls, thus corroborating the possible involvement of mitochondrial dysfunction in the pathophysiology of ADHD.

Although the exact genetic etiology of ADHD is still poorly understood, the contribution of perturbations of mitochondrial performance in the pathophysiology of this disorder must be taken into account [32].

Due to its proximity to the Electron Transport Chain (ETC), mtDNA is highly susceptible to ROS released during oxidative phosphorylation (OXPHOS), and increased oxidative stress could be responsible for direct damage to the mitochondrial genome [33]. In neurodevelopmental disorders, such as ADHD, oxidative stress also has a detrimental effect on mitochondrial energy production in neurons, which are more vulnerable to superoxide anions and hydrogen peroxide [34].

In this situation, the increase in mtDNA content observed in our ADHD cohort might be explained as a defense mechanism against oxidative stress to compensate for the altered oxidative capacity and the defective mitochondrial energy production.

Changes in mtDNA-cn levels could also highlight a link between mitochondrial dysfunction and personality traits of aggressive behavior, an aspect often detected in ADHD [35].

Although data were not statistically significant, in our study, we found a slight difference in mtDNA-cn levels among ADHD patients with different aggressive behaviors, where aggressive ADHD patients exhibited a lower mtDNA-cn content (Figure 4).

Our hypothesis is that this data could be indicative of a more severe mitochondrial impairment among this subgroup of children affected by ADHD.

This interpretation is supported by the evidence that hydrogen peroxide (H_2_O_2_), a ROS by-product originating from the endurance of oxidative stress, could inhibit dopamine release in the striatum, potentially aggravating impulsive and aggressive behavior [36,37]. *MAOA* gene polymorphisms, whose contributions in the development of ADHD and behavioral traits have been widely confirmed, might also be related to mitochondrial dysfunctions [38]. To investigate the existence of a correlation between mitochondrial content and the presence of *MAOA* polymorphisms, we performed a comparison between ADHD patients and healthy controls who shared the same genotypes for both rs6323 and rs1137070 *MAOA* gene polymorphisms.

Our results showed that ADHD patients had a higher mtDNA-cn content compared to healthy controls for all the genotypes of rs6323 and rs1137070 *MAOA* polymorphisms. However, we observed that ADHD patients with the TT genotype of rs6323 and with the CC genotype of rs1137070 had statistically significantly higher mtDNA-cn levels compared to controls carrying the same genotype (Figure 5a,b).

The *MAOA* gene encodes mitochondrial key enzymes essential for the degradation of dopamine, serotonin, and catecholamines. Congenital *MAOA* deficiency, as well as low-activity MAOA variants, has been associated with ADHD and aggressive behavior [39]. This evidence is in line with our observations; also, both genotypes investigated are associated with reduced activity of these outer mitochondrial membrane enzymes [40].

The way in which MAOA metabolizes monoamines leads to the release of ammonia and H_2_O_2_. These degradation byproducts could cross the mitochondrial membrane, causing cardiolipin peroxidation and accumulation of ROS, thus resulting in the inhibition of nuclear Peroxisome Proliferator-Activated Receptor Gamma Coactivator 1 Alpha (PGC-1α) [41]. In this context, the significantly higher mtDNA-cn levels expressed by ADHD patients homozygous for the *MAOA* genotypes, which caused a lower enzymatic activity, might be explained by impaired mitochondrial quality control in a system already compromised by oxidative stress. This condition could cause the fragmentation of mtDNA, which could act as Mitochondrial Damage-Associated Molecular patterns (mtDAMPs), activating the cellular inflammation signaling pathway and culminating in an increase in mitochondrial content.

It has also been demonstrated that the serotonin transporter gene 5-HTT (*SLC6A4*), whose activity in the complex etiology of ADHD has been explored, influences mitochondrial functions and biogenesis [15,42].

For that reason, we investigated whether there was some relationship between the activity of the 5-HTT transporter and mitochondrial content in our cohort carrying different phenotypes of the rs4795541 polymorphism in this gene. We observed that ADHD patients had significantly higher mtDNA-cn levels compared to controls for all the rs4795541 genotypes (LL, SS, LS) (Figure 6a–c).

Moreover, we observed a significant association between mtDNA-cn content and aggressive ADHD patients homozygous (SS) for 5-HTT rs4795541 genotype compared to ADHD aggressive patients carrying the heterozygous (LS) polymorphism (Figure 7). It has been proposed that the SS genotype, which is accountable for a decreased expression of the gene encoding for the 5-HTT transporter, is significantly more prevalent in aggressive children and that it confers more than twice the risk of aggressive behavior [43]. The observation that mtDNA levels were significantly higher in that group of patients might corroborate the hypothesis that mitochondrial dysfunctions are more consistent in these subjects and that the concomitant lower enzymatic activity of the serotonin transporter might confer a greater level of aggressiveness/impulsiveness. On the contrary, heterozygous (LS) polymorphism could partially confer a protective effect against mitochondrial machinery deficiency, as demonstrated by the reduced mtDNA-cn levels observed in this subgroup.

These results are in line with the study performed by Brivio et al., which examined mitochondrial biogenesis in the brains of serotonin transporter (SERT) knockout rats, demonstrating an upregulation of respiratory chain subunits in male rodents [44].

In physiological conditions, mitochondria handle the metabolism of serotonin to guarantee brain homeostasis and correct cognitive outcomes [45]. It has been confirmed that serotonin also plays a crucial role in the regulation of mitochondrial biogenesis in cortical neurons, improving cellular ATP levels and ameliorating both respiratory capacity and OXPHOS efficiency. Moreover, the bond between serotonin and its receptor can trigger Phospholipase C (PCL) and MAPK signaling pathways, resulting in the recruitment of SIRT1, which modulates the expression of PGC-1α, the master regulator of mitochondrial biogenesis [46]. Disturbances in the serotonergic neurotransmission, caused by the interplay between the persistence of oxidative stress and alterations in the serotonin transporter, as in the case of ADHD, could compromise the re-uptake of serotonin in the pre-synapse, thus compromising the maintenance and the regulation of serotonin homeostasis [47]. This situation might possibly affect the SIRT1–PGC-1α axis in the promotion of mitochondrial biogenesis, thus compromising the adaptation in response to altered energetic demands. Furthermore, disturbances in the mitochondrial machinery could negatively affect mitochondrial respiratory chain activity. Indeed, the increase of ROS production, enhanced by the decreased expression of antioxidant enzymes, could lead to progressive mtDNA damage and fragmentation, which is reflected by the higher amount of mtDNA-cn observed in ADHD patients.

This study has several limitations to consider, primarily due to the relatively small sample sizes. However, our strict selection criteria helped minimize potential errors that could arise in larger, more general clinical or population-based surveys. Thus, the effect of population stratification has been reduced as much as possible. Furthermore, these findings are reinforced by multiple testing procedures and, despite applying Bonferroni correction for multiple comparisons, the positive associations remained significant. In addition, unlike randomized controlled trials, case–control studies cannot establish a definitive cause-and-effect relationship but can only identify and measure associations between exposures and outcomes.

Although our findings indicate that mitochondrial alterations may play a role in the pathophysiology of ADHD, further research is necessary to replicate these results across larger populations before definitive conclusions can be drawn.

## 5. Conclusions

Our study highlighted the existence of a correlation between mitochondrial dysfunctions and ADHD, as demonstrated by the significant increase of mtDNA-cn levels in ADHD patients compared to healthy controls.

Although the genetic basis of ADHD is still not fully understood, disturbances in mitochondrial homeostasis, promoted by prolonged oxidative stress and exacerbated by high ROS levels, might be a possible contributing factor in the etiology of that disorder. In this context, cell-free circulating mitochondrial DNA (mtDNA) fragments, released into the bloodstream in response to mitochondrial dysfunction and oxidative damage, have been proposed as promising biomarkers for mitochondrial stress and neurodevelopmental alterations associated with ADHD [48].

These insights underscore the critical role of mitochondrial health in understanding the complex pathophysiology of ADHD and highlight new possibilities for diagnosis and targeted therapeutic interventions.

Moreover, the existing connection between mitochondrial dysfunction and alterations detected in monoaminergic genes and the serotonin transporter might provide a powerful tool to understand the molecular mechanisms that orchestrate the onset of ADHD, offering a novel and innovative therapeutic approach to treat this disease.

## Figures and Tables

**Figure 1 diseases-13-00378-f001:**
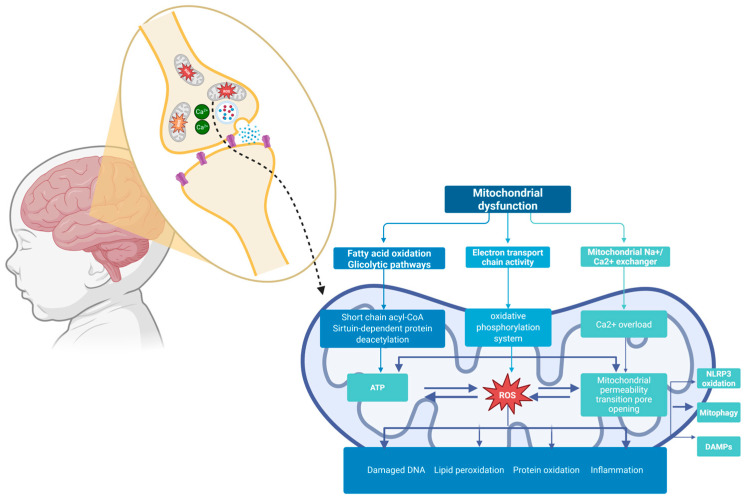
Graphical overview of the mechanisms by which mitochondrial dysfunctions affect the brain in ADHD (figures was created by Biorender.com).

**Figure 2 diseases-13-00378-f002:**
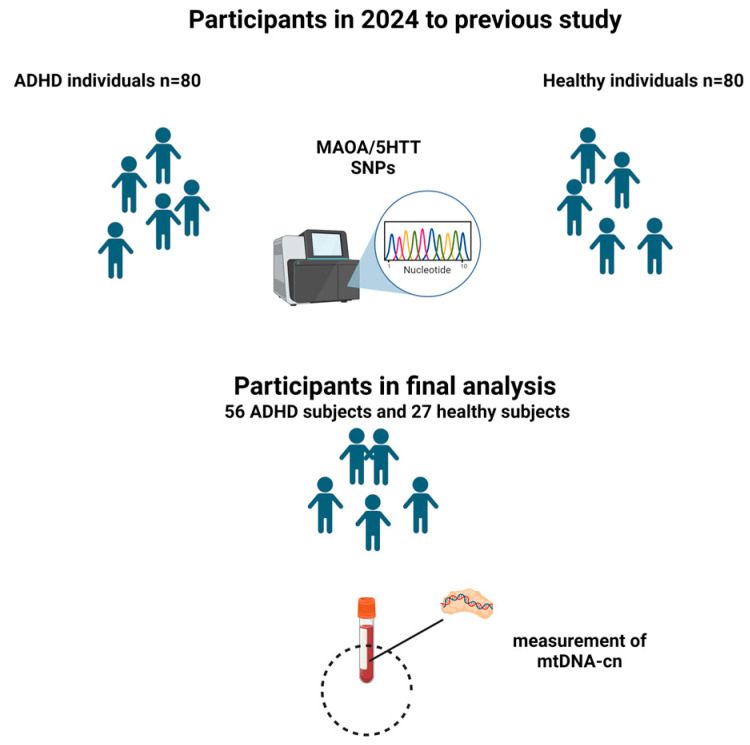
Flowchart of selected subjects for this study. This study was based on a previous research project conducted in 2024 on 160 participants. From the population studied in 2024, 24 participants with ADHD and 53 healthy participants were excluded for poor sample condition, missing mtDNA-cn, or other data. Finally, 56 ADHD individuals and 27 healthy controls were included in this study (figures was created by Biorender.com).

**Figure 3 diseases-13-00378-f003:**
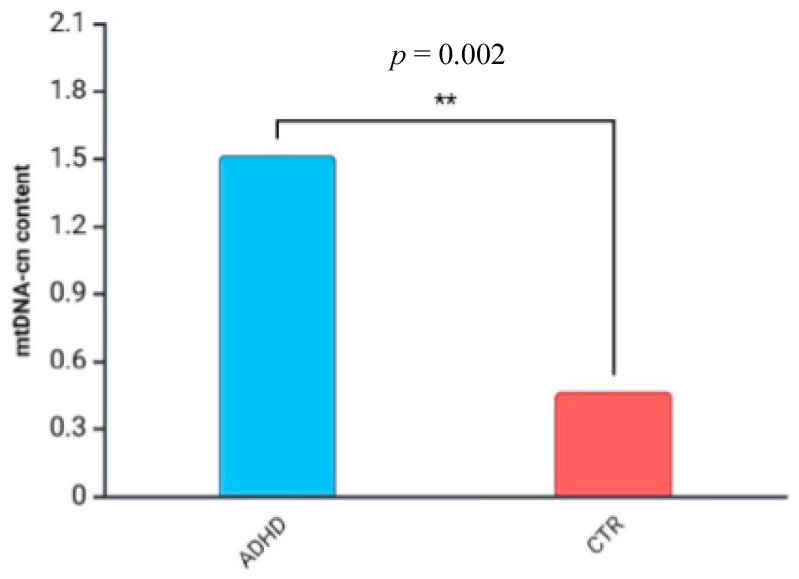
mtDNA-cn content in ADHD patients and healthy controls (asterisks indicate statistical significance with ** *p* < 0.01).

**Figure 4 diseases-13-00378-f004:**
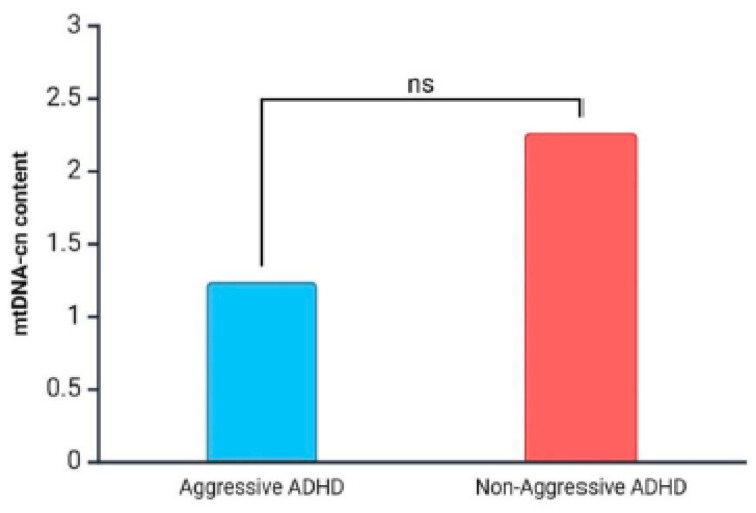
mtDNA-cn content in ADHD patients grouped for aggressive phenotypes (ns: no significant).

**Figure 5 diseases-13-00378-f005:**
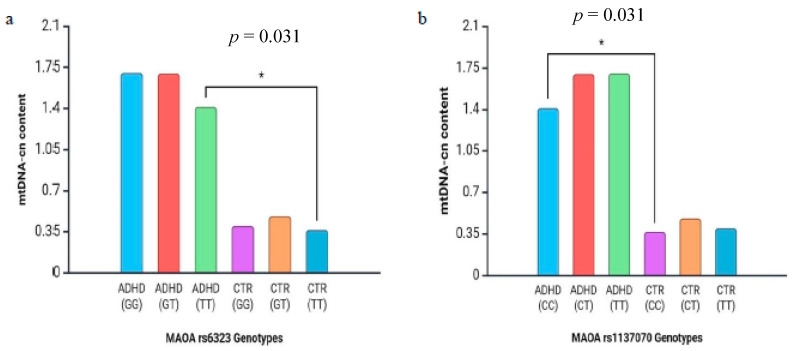
The mtDNA copy number distributions in ADHD patients and control subjects (CTR). *Y*, Y-axis represents the relative mtDNA copy number. (**a**) The mtDNA copy number distributions of patients and controls stratified by *MAOA* rs6323 genotypes. (**b**) The mtDNA copy number distributions of patients and controls stratified by *MAOA* rs1137070 genotypes. *p*, *p*-value was determined by an independent samples *t*-test (asterisks indicate statistical significance with * *p* < 0.05).

**Figure 6 diseases-13-00378-f006:**
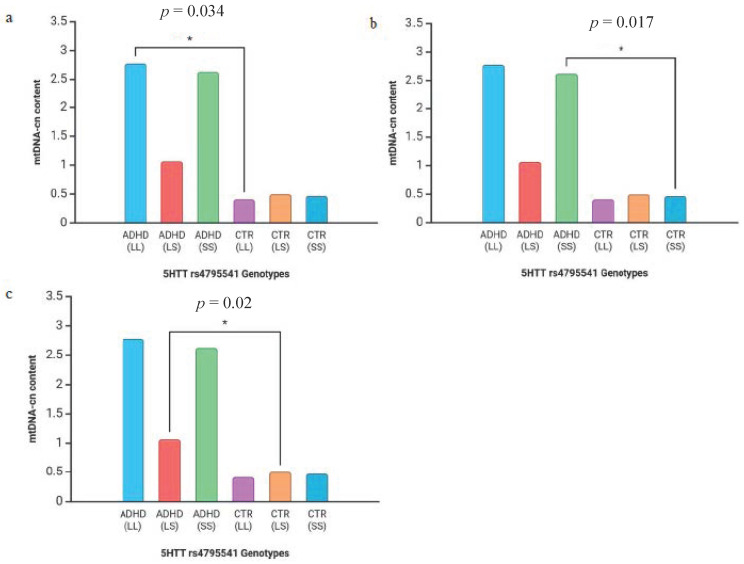
Mitochondrial DNA copy number in the blood of ADHD patients and controls (CTR). The mtDNA copy number distributions of patients and controls have been stratified by 5-HTT rs4795541 LL genotype (**a**), SS genotype (**b**), and LS genotype (**c**) (asterisks indicate statistical significance with * *p* < 0.05).

**Figure 7 diseases-13-00378-f007:**
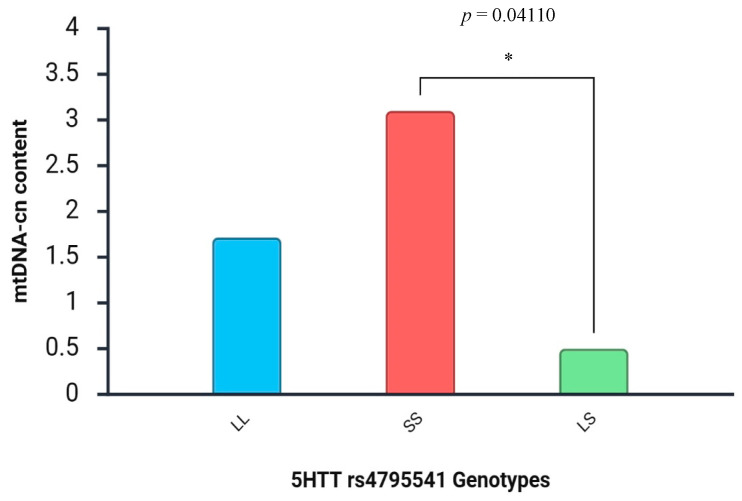
mtDNA copy number distributions in aggressive ADHD patients stratified by 5-HTT rs4795541 genotypes (asterisks indicate statistical significance with * *p* < 0.05).

**Table 1 diseases-13-00378-t001:** Characteristics of ADHD patients.

ADHD Subjects n = 56	
Age (Mean ± SD)	10.1 ± 2.6
Female	2
Male	54
ADHD subtypes	
hyperactive-impulsive	3
inattentive	9
combined	44

**Table 2 diseases-13-00378-t002:** Allele and genotype frequencies of *MAOA* (rs6323 and rs1137070) and 5-HTT (rs4795541) in ADHD individuals and controls. * One sample is not available for rs4795541.

	ADHD		Controls		*p*	OR
MAOA SNP rs6323 (G891T)						
Genotypes	N = 56	%	N = 27	%		
TT	36	0.65	19	0.7	ns	
GT	3	0.05	6	0.23	0.02	0.2
GG	17	0.3	2	0.07	0.01	5.45
Alleles	2n = 112	%	N = 54	%		
T	75	0.67	44	0.81		
G	37	0.33	10	0.19		
MAOA SNP rs1137070 (T1410C)						
Genotypes	N = 56	%	N = 27	%		
CC	36	0.64	19	0.7	ns	
CT	9	0.16	6	0.23	0.0345	0.2
TT	17	0.3	2	0.07	0.0342	4.72
Alleles	2n = 112	%	N = 54	%		
C	81	0.72	44	0.81		
T	43	0.31	10	0.19		
5-HTT SNP rs4795541 (L/S)						
Genotypes	N = 55 *	%	N = 27	%		
LL	14	0.26	5	0.18	ns	
LS	32	0.58	14	0.52	ns	
SS	9	0.16	8	0.3	ns	
Alleles	2n = 110 *	%	N = 54	%		
L	*60*	0.54	*24*	0.45		
S	*50*	0.46	*30*	0.55		

## Data Availability

The data presented in this study are available on request from the corresponding author.

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
