# Peer review of "Exploring Mitochondrial DNA Copy Number in Italian Children with ADHD: Implications for Neurobiological Mechanisms"

_diseases, 2025, doi:10.3390/diseases13110378_

Round 1
Reviewer 1 Report
Comments and Suggestions for Authors
This manuscript aims to investigate the potential contribution of mitochondrial dysfunctions to the etiology of ADHD. This topic appears scientifically relevant and of interest to a broad readership due to its novelty, depth, and presentation. The paper has been well-established, providing suggestions that changes in mtDNA-cn could mirror disturbances in mitochondrial homeostasis and exacerbated oxidative stress, but it needs some small improvements to make it more useble.
The introduction provides an appropriate background which clearly explains why authors have been focused on that topic. The premises elucidate the relationship between ADHD and mitochondrial activity in terms of energy production, metabolism, and cellular signaling. Moreover, the importance of mitochondrial DNA copy number as a marker for mitochondrial health and disease is well articulated.
However, authors should improve the writing skills of the manuscript, checking for grammatical mistakes or tense inconsistencies.
-Gene nomenclature needs to be fixed (italicized) (e.g. MAOA, SLC6A4) in the main text, following the most recent HUGO approved names.
-Some typos that need to be corrected (e.g. H2O2) in the discussion section.
Although the small sample size of the cohort investigated, the research design is comprehensive, well-organized, and it well focuses on the topic presented by the authors. Inclusion and exclusion criteria applied for patient selection have been described in detail and the comprehensive approach applied conveys a more detailed understanding of the main elements underlying this neurodevelopmental disorder.
The methods are adequately described, and the experimental approach is in line with the purpose of the investigations. When authors state that mtDNA-cn levels have been assessed by quantitative Real-Time PCR, they should specify the model of the Instrument used to perform the experiments.
The conclusions support the results obtained from authors, also highlighting the broader implications, limitations, and future directions. Furthermore, in this section authors exhaustively explain the reason why some biological mechanisms underlying ADHD and ascribable to mitochondrial dysfunctions (e.g. high ROS levels and oxidative stress) might reflect underlying alterations in neurochemical metabolism.
Figures and tables are clearly understandable. Since table 2 reports a large amount of detailed numerical and textual data, it should be preferrable to split tables 1 and 2 into two.
Author Response
This manuscript aims to investigate the potential contribution of mitochondrial dysfunctions to the etiology of ADHD. This topic appears scientifically relevant and of interest to a broad readership due to its novelty, depth, and presentation. The paper has been well-established, providing suggestions that changes in mtDNA-cn could mirror disturbances in mitochondrial homeostasis and exacerbated oxidative stress, but it needs some small improvements to make it more usable.
The introduction provides an appropriate background which clearly explains why authors have been focused on that topic. The premises elucidate the relationship between ADHD and mitochondrial activity in terms of energy production, metabolism, and cellular signaling. Moreover, the importance of mitochondrial DNA copy number as a marker for mitochondrial health and disease is well articulated.
However, authors should improve the writing skills of the manuscript, checking for grammatical mistakes or tense inconsistencies.
-Gene nomenclature needs to be fixed (italicized) (e.g. MAOA, SLC6A4) in the main text, following the most recent HUGO approved names.
Reply: We apologies for the mistakes performed on gene nomenclature, we provided to check and to fix the name of the genes in the main manuscript.
-Some typos that need to be corrected (e.g. H2O2) in the discussion section.
Reply: We are grateful to the reviewer for pointing out the typos. We performed a careful revision of the manuscript and we corrected all the typing errors.
Although the small sample size of the cohort investigated, the research design is comprehensive, well-organized, and it well focuses on the topic presented by the authors. Inclusion and exclusion criteria applied for patient selection have been described in detail and the comprehensive approach applied conveys a more detailed understanding of the main elements underlying this neurodevelopmental disorder.
-The methods are adequately described, and the experimental approach is in line with the purpose of the investigations. When authors state that mtDNA-cn levels have been assessed by quantitative Real-Time PCR, they should specify the model of the Instrument used to perform the experiments.
Reply: We are grateful to the reviewer for noticing this omission. We added the name of the Real-Time PCR instrument used to perform the experiments in the method section of the manuscript.
The conclusions support the results obtained from authors, also highlighting the broader implications, limitations, and future directions. Furthermore, in this section authors exhaustively explain the reason why some biological mechanisms underlying ADHD and ascribable to mitochondrial dysfunctions (e.g. high ROS levels and oxidative stress) might reflect underlying alterations in neurochemical metabolism.
-Figures and tables are clearly understandable. Since table 2 reports a large amount of detailed numerical and textual data, it should be preferable to split tables 1 and 2 into two.
Reply: We recognize that the way table 2 is split makes it difficult to read. We provided to insert table 2 in a single page, hoping that in this way data are clearly understandable.
Reviewer 2 Report
Comments and Suggestions for Authors
- i have major concern about sex of the children involved. sex is an important covariate for mitochondrial dna copy number. females typically show higher mtdna-cn than males due to estrogen-mediated stimulation of mitochondrial biogenesis x-chromosome effects on mitochondrial genes and sex-specific metabolic demands. however with only 2 females among 56 adhd patients your study cannot adequately control for or assess sex effects. the results should be explicitly presented as male-specific findings with the severe gender imbalance acknowledged as a critical limitation that prevents generalization to females with adhd.
- to solve the above re-analysis is needed with boys only. the paper need to be also pilot.
- pls also clarify how controls were matched and verify they underwent equivalent screening to rule out subclinical adhd symptoms or psychiatric conditions.
- in methods author reported the age at diagnosis ranged from 6 to 17 years. then they mentioned to evaluate the trend in mtdna content across different age groups in adhd patients, participants were divided into two categories: 6–11 years and 12–18 years. is age 6-17 or 6-18? the 6-11 vs 12-18 age comparison shows high variance and no significance. either expand the sample or remove this speculative analysis.
- for adhd the conners' provocation/aggressivity subscale alone is insufficient. pls state who and how and which criteria dsm/icd were used for adhd classification and diagnosis.
- the relationship between specific genotypes and aggressive behavior within adhd subgroups is unexplored. add this analysis or acknowledge as a limitation.
- language need to be toned down for smal scale study/.
Author Response
I have major concern about sex of the children involved. sex is an important covariate for mitochondrial dna copy number. Females typically show higher mtdna-cn than males due to estrogen-mediated stimulation of mitochondrial biogenesis x-chromosome effects on mitochondrial genes and sex-specific metabolic demands. However, with only 2 females among 56 adhd patients your study cannot adequately control for or assess sex effects. the results should be explicitly presented as male-specific findings with the severe gender imbalance acknowledged as a critical limitation that prevents generalization to females with adhd. To solve the above re-analysis is needed with boys only. the paper need to be also pilot.
Reply: We greatly appreciate the reviewer's opinion and certainly agree with the main concerns. As reported in the main text, we acknowledged and listed the several limitations present in our study. As known, ADHD has a higher prevalence in males, and this gender gap could undoubtedly represent a restraint for the outcomes obtained in this study. However, as explained in the manuscript, our findings are reinforced by multiple testing procedures and, despite applying Bonferroni correction for multiple comparisons, the positive associations among ADHD and mtDNA-cn levels remained significant. Moreover, we did not perform any analysis that encompassed a correlation between mtDNA-cn levels and sex. A larger and more homogeneous population will certainly be needed to draw any conclusions in this regard. In our opinion, the analysis conducted should remain unchanged also to highlight the dissimilar prevalence of the disease in our small Italian cohort.
-pls also clarify how controls were matched and verify they underwent equivalent screening to rule out subclinical adhd symptoms or psychiatric conditions.
Reply: We thank the reviewer for this suggestion. In our study healthy controls underwent to the same evaluations of ADHD patients before the effective recruitment and we provided to clarify this in the main text.
-in methods author reported the age at diagnosis ranged from 6 to 17 years. Then they mentioned to evaluate the trend in mtdna content across different age groups in adhd patients, participants were divided into two categories: 6–11 years and 12–18 years. is age 6-17 or 6-18?
Reply: We apologize with the reviewer for the typo in the main text. The age range of our cohort was 6- 18 years and we provided to correct this mistake in the methods section.
-the 6-11 vs 12-18 age comparison shows high variance and no significance. either expand the sample or remove this speculative analysis.
Reply: We agree with the suggestion of the reviewer and we decided to remove this speculative analysis from our manuscript.
-for adhd the conners' provocation/aggressivity subscale alone is insufficient. pls state who and how and which criteria dsm/icd were used for adhd classification and diagnosis.
Reply: We are grateful to the reviewer for this suggestion. We have expanded the details of the diagnostic criteria applied in our study, for a more accurate screening of ADHD patients.
-the relationship between specific genotypes and aggressive behavior within adhd subgroups is unexplored. add this analysis or acknowledge as a limitation.
Reply: We appreciate the suggestion of the reviewer. We performed this further analysis and we reported our results in the main text.
-language need to be toned down for small scale study/.
Reply: We thank the reviewer for this recommendation. As suggested we stated that, although promising, the results of our work are still preliminary and that further research is necessary to replicate these results across larger populations.
Round 2
Reviewer 2 Report
Comments and Suggestions for Authors
dear colleagues,
my comments were not addressed and study ratio of males to females is 27:1
this is not not justified nor acceptable. in general, females have a higher mitochondrial DNA copy number (mtDNA-cn) than males. studies have shown that males have significantly lower mtDNA-cn in their blood compared to females across various age groups.
acceptable ratio will be 4-3:1 we will also accept 5-6:1 but this data is mainly boys
fewer controls than cases could limit the ability to draw robust conclusions from the study. analysis need to be done matching cases and controls to ensure conclusions are made.
Author Response
We appreciate the suggestion of the reviewer and we understand the main concern. Therefore, for a deeper and accurate evaluation of our data, we performed an odds ratio analysis stratified by sex to evaluate the association between mtDNA-CN levels and ADHD risk. Our results showed that the risk remains high and statistically significant, both in the total group (male and female) and in the males group alone. We also recognized that ADHD has a higher prevalence in males, so it could be difficult to obtain a homogeneous cohort to examine. However, this should not affect our findings, particularly since no correlation between mtDNA-CN levels and sex was assessed. Our hypothesis is also supported by current literature in that topic, which clearly report that fluctuations in mitochondrial genomic content could be related to a particular disease, without evidence of sex-dependent effects.
Round 3
Reviewer 2 Report
Comments and Suggestions for Authors
no more comments